# Characterization of Beta-Lactam Resistome of *Escherichia coli* Causing Nosocomial Infections

**DOI:** 10.3390/antibiotics12091355

**Published:** 2023-08-23

**Authors:** Rosalino Vázquez-López, Tanya Hernández-Martínez, Selene Ivonne Larios-Fernández, Celia Piña-Leyva, Manuel Lara-Lozano, Tayde Guerrero-González, Javier Martínez-Bautista, Eduardo Gómez-Conde, Juan Antonio González-Barrios

**Affiliations:** 1Departamento de Microbiología, Centro de Investigación en Ciencias de la Salud (CICSA), Facultad de Ciencias de la Salud Universidad Anáhuac México Norte, Huixquilucan 52786, Mexico; rosalino.vazquez@anahuac.mx; 2Laboratorio de Medicina Genómica, Hospital Regional “Primero de Octubre”, ISSSTE, Av. Instituto Politécnico Nacional 1669, Lindavista, Gustavo A. Madero, Ciudad de México 07300, Mexico; aynat_hm@hotmail.com (T.H.-M.); silf.b@hotmail.com (S.I.L.-F.); plceliaqfb62812@gmail.com (C.P.-L.); manuellara.mvz@gmail.com (M.L.-L.); taygercita.19@gmail.com (T.G.-G.); 3Laboratorio de Microbiología, Hospital Regional “Primero de Octubre”, ISSSTE, Av. Instituto Politécnico Nacional 1669, Lindavista, Gustavo A. Madero, Ciudad de México 07300, Mexico; javier.martinezb@issste.gob.mx; 4Departamento de Inmunobiología, Facultad de Medicina, Benemérita Universidad Autónoma de Puebla (BUAP), Puebla 72420, Mexico; eduardo.gomezc@imss.gob.mx

**Keywords:** *Escherichia coli*, beta-lactam resistome, antibiotic multi-resistance, pathotypes

## Abstract

Nosocomial infections caused by *Escherichia coli* pose significant therapeutic challenges due to the high expression of genes encoding antimicrobial drug resistance. In this study, we investigated the conformation of the beta-lactam resistome responsible for the specific pattern of resistance against beta-lactam antibiotics. A total of 218 *Escherichia coli* strains were isolated from in-hospital patients diagnosed with nosocomial infections, obtained from various sources such as urine (*n* = 49, 22.48%), vaginal discharge (*n* = 46, 21.10%), catheter tips (*n* = 14, 6.42%), blood (*n* = 13, 5.96%), feces (*n* = 12, 5.50%), sputum (*n* = 11, 5.05%), biopsies (*n* = 8, 3.67%), cerebrospinal fluid (*n* = 2, 0.92%) and other unspecified discharges (*n* = 63, 28.90%). To characterize the beta-lactam resistome, all strains were subjected to antibiotic dilution tests and grown in beta-lactam antibiotics supplemented with Luria culture medium. Subsequently, multiplex PCR and next-generation sequencing were conducted. The results show a multi-drug-resistance phenotype, particularly against beta-lactam drugs. The primary determinant of this resistance was the expression of the blaTEM gene family, with 209 positive strains (95.87%) expressing it as a single gene (*n* = 47, 21.6%) or in combination with other genes. Common combinations included *bla*TEM + *bla*CTX (*n* = 42, 19.3%), *bla*TEM + *bla*CTX + *bla*SHV (*n* = 13, 6%) and *bla*TEM + *bla*CTX + *bla*BIL (*n* = 12, 5.5%), among others. The beta-lactam resistome of nosocomial *Escherichia coli* strains isolated from inpatients at the “October first” Regional Hospital of ISSSTE was predominantly composed of members of the blaTEM gene family, expressed in various configurations along with different members of other beta-lactamase gene families.

## 1. Introduction

In the past two decades, nosocomial infections have become a significant public health problem worldwide [1], representing a healthcare failure that leads to adverse effects during in-hospital stays and an increased risk of mortality among patients [2]. These infections are associated with various medical procedures such as invasive interventions, extensive surgeries and the use of indwelling medical devices, giving rise to central-line-associated bloodstream infections (CLABSIs), catheter-associated urinary-tract infections (CAUTIs), surgical-site infections (SSIs) and ventilator-associated pneumonia (VAP) [3].

Factors such as prolonged hospital stays also contribute to the acquisition of bloodstream infections not associated with central catheter use, gastrointestinal infections (GIs), urinary-tract infections (UTIs) not associated with catheter use and non-ventilator-associated hospital-acquired pneumonia (NV-HAP). Various body sites are affected by healthcare-associated infections (HAIs), including the ear, eye, nose, throat, lower respiratory tract, skin, soft tissue, cardiovascular system, bone, joints, central nervous system and reproductive tract. HAIs have diverse causes, including bacterial, viral or fungal agents. Among the main bacteria responsible for these infections are Gram-positive microorganisms like coagulase-negative Staphylococci, *Staphylococcus aureus, Streptococcus species, Enterococcus faecalis, Enterococcus faecium* and *Clostridium difficile* [4], as well as Gram-negative microorganisms such as some members of the Enterobacteriaceae family (e.g., *Escherichia coli, Klebsiella pneumoniae, Klebsiella oxytoca, Proteus mirabilis*), Enterobacter species (*Pseudomonas aeruginosa, Acinetobacter baumanii* and *Burkholderia cepacian*). These pathogens are associated with high morbidity and mortality rates in hospitalized patients due to their inherent multi-drug-resistant properties [5].

The extensive use of antibiotics in hospitals exerts significant selection pressure on nosocomial bacterial populations, especially on bacteria like *Escherichia coli, Acinetobacter baumanii, Pseudomonas aeruginosa, Enterococcus faecium, Proteus mirabilis* and *Stenotrophomonas maltophilia*, leading to the emergence of resistant, multi-resistant and pan-resistant nosocomial infections [6]. *Escherichia coli*, a pathogenic microorganism, can be found in various environmental sources and the intestines of humans and animals. While most strains are harmless, certain pathogenic strains have been associated with gastrointestinal diseases and extraintestinal tissue infections. In hospitals, *Escherichia coli* has been identified as the causative agent of various healthcare-associated infections like BSI, CLABSI, UTI, CAUTI, SSI, VAP, NV-HAP and GI [7].

Due to excessive and sometimes indiscriminate antibiotic use, *Escherichia coli* strains in hospitals have become highly multi-drug-resistant (MDR). This resistance is particularly common in beta-lactam antibiotics, severely limiting effective treatment options [8]. The resistance to beta-lactam antibiotics is encoded by different classes of beta-lactamases, including *bla*CTX, *bla*VIM, *bla*NMC, *bla*OXA, *bla*IMP, *bla*LAP, *bla*ROB, *bla*DHA, *bla*VEB, *bla*CMY, *bla*LAP, *bla*SHV, *bla*TEM, *bla*SME, *bla*SFC, *bla*KPC, *bla*AmpC, *bla*GES, *bla*SME, *bla*PER and *bla*NDM [9,10,11,12,13,14,15,16,17,18], which confer a wide spectrum of resistance to beta-lactam antibiotics [19].

In this study, we focus on characterizing the beta-lactam resistome expressed by *Escherichia coli*-causing nosocomial infections in the “October first” Regional Hospital of ISSSTE in Mexico.

## 2. Materials and Methods

### 2.1. Primer Design

The bioinformatic analysis and primer design process were previously described and reported by our research group [20]. In summary, we retrieved beta-lactamase integron and DNA sequences from the GenBank of NCBI. All available sequences of beta-lactamases were gathered for the purpose of PCR primer design. The DNA sequences were aligned using ClustalW v.2 software [21]. Subsequently, we used FigTree V1.4.0 software [22] to generate phylogenetic trees based on the alignments. From the alignment results, conserved sequences were identified and used as a basis for designing a set of specific and degenerate primers. The PerPrimer v1.1.21 Software [23] was utilized, employing stringent criteria, including primer length (18–25 bp), a melting temperature (Tm) of 60–62 ${}^{\circ }$C, a GC content of 40–60%, with a ΔT${}^{\circ }$ of 1 ${}^{\circ }$C in the annealing temperature and an amplicon size ranging from 83 to 230 bp [20] (Table 1).

### 2.2. Biological Samples

A total of five-hundred-and-eighty-five clinical samples were collected from various secretions, including urethral, vaginal, nasal, diabetic foot ulcers, bronchial, surgical and pharyngeal wound secretions. Additionally, body fluids such as blood, dialysis fluid, cerebrospinal fluid, peritoneal, pleural and synovial fluid, urine, feces, sputum and semen, as well as catheter tips from patients diagnosed with nosocomial infections, were also included in the study. Among these samples, 218 *Escherichia coli* strains showing multi-drug resistance were cultured and selected for further analysis. The biochemical identification of these strains was performed by the Microbiology section of the Central Clinical Laboratory at the “October first” Regional Hospital (ISSSTE).

### 2.3. *Escherichia coli* Isolation Culture

This study was conducted at the “October first” Regional Hospital (ISSSTE) in Mexico City. The collected samples were cultured on various agar media, including Blood Agar (COS), MacConkey Agar, Chocolate Mueller Hilton Agar (PVX), Chromogenic Candida Agar, Sabouraud Agar, Triptone Soya Broth, Chromogenic Agar for *Staphylococcus aureus* identification in human specimens (SAID), Salted Mannitol Agar, Chromogenic Agar for the detection of Salmonella, SS Agar for Shigella and Salmonella and CromlDTM CPS Agar.

The cultured plates were incubated at 35 ${}^{\circ }$C for 18 to 24 h. All bacterial growth on the MacConkey agar plates was subjected to biochemical identification tests. However, only the cultures that were identified as *Escherichia coli* were further selected for antimicrobial susceptibility testing and genotyping.

### 2.4. Biochemical Identification

The biochemical identification of *Escherichia coli* strains was carried out using the VITEK 2XL system, an automated method provided by bioMérieux Inc. (Durham, NC, USA) Specifically, the VITEK 2XL GN 21341 cards were used, containing 21 biochemical tests. These tests included β-galactosidase (OPNG), arginine dehydrolase (ADH), lysine decarboxylase (LDC), ornithine decarboxylase (ODC), citrate utilization (CIT), hydrogen sulfide (H_2_S) production, urease (URE) activity, tryptophan deaminase (TDA) production, indole production (IND), Voges–Proskauer acetoin production (VP), gelatinase (GEL) activity and fermentation/oxidation tests for various sugars such as glucose (GLU), mannitol, inositol, sorbitol, ramnose, sucrose, melibiose, amygdalin (vitamin B17), arabinose and cytochrome oxidase (OX) activity.

To confirm the *Escherichia coli* identification, the strains must test positive for β-galactosidase (OPNG), lysine decarboxylase (LDL), ornithine decarboxylase (ODC), indole production (IND), glucose fermentation (GLU), mannitol (MAN), sorbitol (SOR), ramnose (RHA), sucrose (SAC), melibiose (MEL) and arabinose (ARA) tests among the set of biochemical tests conducted using the VITEK 2XL GN 21341 cards.

### 2.5. Antimicrobial Susceptibility Testing

Antibiotic susceptibility testing was performed using the VITEK 2XL automated system with ASTGN70 cards (REF 413 and 401) specifically designed for Gram-negative bacteria by BioMérieux Inc.(Durham, NC, USA) The following antibiotics were tested with varying concentrations (in μg/mL): amikacin (8, 16, 64 μg/mL), ampicillin (4, 8, 32 μg/mL), ampicillin/sulbactam (4/2, 16/8, 32/16 μg/mL), aztreonam (2, 8, 32 μg/mL), cefazolin (4, 16, 64 μg/mL), cefepime (2, 8, 16, 32 μg/mL), cefotetan (2, 8, 32 μg/mL), ceftazidime (1, 2, 8, 32 μg/mL), ceftriaxone (1, 2, 8, 32 μg/mL), cefuroxime (2, 8, 32 μg/mL), ciprofloxacin (0.5, 2, 4 μg/mL), gentamicin (4, 16, 32 μg/mL), imipenem (2, 4, 16μg/mL), meropenem (0.5, 4, 16 μg/mL), piperacillin/tazobactam (4/4, 16/4, 128/4 μg/mL), ticarcillin/clavulanic acid (8/2, 32/2, 64/2 μg/mL), tobramycin (8, 16, 64 μg/mL), trimethoprim/sulfamethoxazole (0.5/9.5, 2/38, 16/304 μg/mL), nitrofurantoin (16, 32, 64 μg/mL) and gatifloxacin (1, 4, 8 μg/mL).

The resistance or susceptibility profile of each strain was determined by measuring the minimum inhibitory concentration (MIC) of the antibiotics, which was assessed by measuring the optical density of bacterial cultures after incubation.

### 2.6. Beta-Lactam Selection

For genomic and plasmid DNA sequencing, it was necessary to obtain generic cultures of isolated pure *Escherichia coli* colonies. The growth of each colony was achieved by placing it in 10 mL of 25% lysogeny broth (LB) medium, which was then sterilized at 121 ${}^{\circ }$C for 15 min. Subsequently, each *Escherichia coli* strain was grown in a set of beta-lactam drugs, including ampicillin (10 μg), ampicillin/sulbactam (10/10 μg), carbenicillin (100 μg), cefaclor (30 μg), cefazolin (30 μg), cefepime (30 μg), cefoperazone (75 μg), cefotetan (30 μg), mezlocillin (75 μg) and piperacillin/tazobactam (100/10 μg). All of these beta-lactam drugs were obtained from Becton Dickinson (Franklin Lakes, NJ, USA). Each medium was incubated at 37 ${}^{\circ }$C for 24 h under aerobic conditions. The inoculum was adjusted using a Mac Farland 0.5 reading, which corresponds to an expected colony-forming units per milliliter (CFU/mL) of approximately 1.5 × 108. The cultures were then placed in a Shaker Series Innova at 37 ${}^{\circ }$C and 240 rpm for 20 h.

### 2.7. DNA Extraction

#### 2.7.1. Crude Extract

Total DNA was extracted using a modified version of the heat-shock method described by Maugeri et al. One milliliter of Luria medium containing a well-identified, multi-resistant *Escherichia coli* strain grown for 20 h at 35 ${}^{\circ }$C was used. The selected medium was lysed by heating it in a Thermoblock for 1 min at 100 ${}^{\circ }$C, followed by immediate placement in an ice bath for 5 min, inducing thermal shock and facilitating the release of genetic material. Two milliliters of the cell lysates containing DNA were purified using chelex (glass beads). The purified DNA served as a template for both real-time PCR assays and NGS (next-generation sequencing). The extracted DNA was stored at −80 ${}^{\circ }$C for further analysis and experimentation.

#### 2.7.2. Plasmid DNA Extraction

Plasmid DNA extraction was conducted using a previously reported method [20]. The PureLink HiPure Plasmid DNA Purification Kit (Invitrogen, Carlsbad, CA, USA) was employed, following the manufacturer’s instructions. The extracted plasmid DNA solution was quantified using a Qubit Flex Fluorometer (Thermo Fisher Scientific, Wilmington, DE, USA) based on fluorometry. To verify the integrity of the plasmid DNA, electrophoresis was performed in 2% agarose gel. Subsequently, the samples were stored at −80 ${}^{\circ }$C until further use.

### 2.8. Real-Time PCR

The beta-lactamases SYBR Green-based qPCR assays were performed on the Real-Time HT FAST 7900 system (Applied Biosystems, Waltham, MA, USA). Each reaction was conducted in a total volume of 10 μL, comprising 5 μL of 21.5 × SYBR Green PCR Master Mix (Applied Biosystems, Waltham, MA, USA), 10 mM of specific or degenerate primers and 3 μL of the DNA template. The cycling conditions for both assays consisted of an initial denaturation step at 95 ${}^{\circ }$C for 5 min, followed by 45 cycles of denaturation at 95 ${}^{\circ }$C for 15 s, annealing at 50 ${}^{\circ }$C for 40 s and extension at 60 ${}^{\circ }$C for 40 s. To confirm the specificity of the qPCR products, the amplicons were analyzed using melting curves. Additionally, the products were visualized on a 2% agarose gel pre-stained with ethidium bromide and the resulting image was digitized using a GelLogistic 3000 photodocumenter.

### 2.9. Whole DNA Sequencing

For DNA sequencing, a previously reported method by the research group [20] was followed. Both genomic and plasmid DNA were mixed in a 1:1 proportion and used to prepare indexed libraries using the Illumina Nextera XT DNA Sample Preparation Kit (FC-131-1096) designed for small genomes. The libraries were then sequenced on the MiSeq platform (Illumina; San Diego, CA, USA). Adapters and barcodes were trimmed using the default settings in the Illumina experiment manager, resulting in 300 bp paired-end reads. The quality of the unprocessed reads was assessed using FastQC High-Throughput Sequence QC Report v:0.11.5 from Babraham Bioinformatics, Babraham Institute (Cambridge, UK) [24]. A minimum quality score (Q score) of more than 30 was achieved for at least 85% of all reads. For read mapping, the BWA-MEM aligner version 0.7.7-r441 from the Wellcome Trust Sanger Institute (Hinxton, UK) was used. Genome assembly was performed using the SPAdes Genome Assembler software version 3.11 from CAB, St. Petersburg State University (St. Petersburg, Russia). The reported ASM74325v1 genome of Escherichia coli strain 25922 ATCC with a size of 5.2 Mbp was used as the assembly reference [25]. The DNA metagenomic analysis for the taxonomic classification of the Escherichia coli bacteria was conducted using the Kraken taxonomic sequence classification system, version 0.10.5-beta, from CCB, Johns Hopkins University (Baltimore, MD, USA) [26]. For identifying beta-lactamase genes, a comparative analysis was performed using the Basic Local Alignment Search Tool (BLAST) from NCBI-NIH (Bethesda, MD, USA) [27].

### 2.10. Study Outcomes

The main objective of this study was to characterize the beta-lactam resistome expressed by *Escherichia coli* strains causing nosocomial infections. The focus was on understanding the patterns of antibiotic resistance induced by the expression of beta-lactamases, which often complicate the treatment of nosocomial infections. By examining the resistome and identifying the specific beta-lactamase genes responsible for the multi-drug resistance, this study aimed to provide valuable insights for the better management and treatment of nosocomial infections caused by these bacteria.

## 3. Results

### 3.1. Sampling Sources

We analyzed 585 clinical samples from various anatomopathological sources of in-hospital patients with nosocomial infections at the “October first” Regional Hospital in Mexico City. Of these, we isolated 218 multi-resistant *Escherichia coli* strains, with 63 (28.90%) being unspecified discharge, 49 (22.48%) from urine cultures, 46 (21.10%) from vaginal discharge, 14 (6.42%) from catheter tips, 13 (5.96%) from blood cultures, 12 (5.50%) from feces cultures, 11 (5.05%) from sputum, 8 (3.67%) from biopsies and 2 (0.92%) from cerebrospinal fluid.

The microbiology laboratory reported that all 218 strains were multi-drug-resistant, showing 100% resistance to ampicillin, ampicillin/sulbactam, cefazolin, ceftriaxone and cefepime. Additionally, 8.26% (*n* = 18) were resistant to ertapenem, 28.34% (*n* = 13) to meropenem, 58.26% (*n* = 217) to gentamicin, 50.00% (*n* = 109) to tobramycin, 91.28% (*n* = 199) to ciprofloxacin and 16.51% (*n* = 36) to nitrofurantoin. However, no resistance was found for amikasin and tigecycline (Table 2).

### 3.2. Multiple-Antibiotic-Resistance Index (MAR Index)

We calculated the multiple-antibiotic-resistance index (MAR index) for 218 *Escherichia coli* strains and found that 216 strains had an MAR index greater than 0.25. Among these, 167 strains had an MAR index ranging from 0.5 to 0.75. We did not find any strains with an MAR index of 0.05 to 0.2 or 0.95 to 1.0. The MAR index for all nosocomial *Escherichia coli* isolates was determined to be 0.57 ± 0.21 (Table 3).

### 3.3. Beta-Lactam Selection

We tested the *Escherichia coli* subculture in liquid beta-lactam selection medium and found that all 218 strains showed growth in medium supplemented with ampicillin, cefazolin and cefazolin. Furthermore, 87.16% (*n* = 190) of strains showed growth in amoxicillin/clavulanic acid, 85.32% (*n* = 190) in cefixime, 72.48% (*n* = 158) in cefuroxime, 71.10% (*n* = 155) in cephalothin, 67.43% (*n* = 158) in cefaclor, 64.68% (*n* = 141) in piperacillin, 63.30% (*n* = 138) in cefazolin and 52.29% (*n* = 114) in doxycycline. The lower resistance frequencies were 17.89% (*n* = 39) to piperacillin, 15.60% (*n* = 39) to ceftizoxime, 13.76% (*n* = 30) to meropenem and 11.01% (*n* = 24) to imipenem (Table 2). These results yielded a specific beta-lactam MAR index of 0.82 ± 0.09 (Table 4).

### 3.4. Sequencing *Escherichia coli* Identification and Genome Size

The automatic identification of 218 different strains of *Escherichia coli* was confirmed by performing whole-DNA sequencing using NGS (Table 4). The results showed 100% concordance with the automated bacterial identification. The genome sequence analysis revealed a wide diversity in the genome sizes of the *Escherichia coli* strains, with a standard genome size of 5,052,750 ± 218,517 bp. The minimum genome size was 4,566,280 bp and the maximum was 6,070,710 bp. The mode and median of genome size were 4,876,890 bp and 5,048,520 bp, respectively (Figure 1).

### 3.5. qPCR for Beta-Lactamases

The qPCR enabled the identification of beta-lactamase families, including *bla*TEM, *bla*CTX, *bla*SHV, *bla*BIL, *bla*DHA, *bla*IMP, *bla*LAP, *bla*P, *bla*VIM, *bla*KPC, *bla*CMY, *bla*OXA and *bla*ROB (Figure 2).

### 3.6. Beta-Lactamases Gene Family Identification

The qPCR results revealed that all 218 *Escherichia coli* strains carried at least one beta-lactamase gene. We identified the transcribed gene families and the phenotype responsible for beta-lactam drug resistance. Specifically, 209 strains (95.87%) were positive for *bla*aTEM, 155 (71.10%) for *bla*CTX, 74 (33.94%) for *bla*SHV, 65 (29.82%) for *bla*BIL, 42 (19.27%) for *bla*DHA, 41 (18.81%) for *bla*CMY, 24 (11.01%) for *bla*IMP, 15 (6.88%) for *bla*LAP, 10 (14.59%) for *bla*P, 9 (4.13%) for *bla*VIM, 4 (1.82%) for *bla*CTX and 2 (0.92%) for *bla*CTX (Table 5).

### 3.7. Beta-Lactamase Phenotypes

Of the 218 cultures that tested positive for beta-lactam gene families, 84% of the strains carried the most common beta-lactam resistomes (Table 6). Specifically, 21.56% (*n* = 47) of strains carried only the *bla*TEM phenotype, 23.39% (*n* = 51) carried two different gene families, including the *bla*TEM + *bla*CTX, *bla*SHV + *bla*DHA, *bla*TEM + *bla*CMY, *bla*TEM + *bla*LAP, *bla*CTX + *bla*BIL and *bla*CTX + *bla*VIM phenotypes. In total, 29.36% (*n* = 64) carried three members of beta-lactamase gene families, including the *bla*TEM + *bla*SHV + *bla*CTX, *bla*TEM + *bla*CTX + *bla*BIL, *bla*aTEM + *bla*CTX + *bla*DHA, *bla*TEM + *bla*CTX + *bla*CMY, *bla*SHV + *bla*CTX + *bla*BIL, *bla*TEM + *bla*SHV + *bla*BIL, *bla*TEM + *bla*CTX + *bla*VIM, *bla*TEM + *bla*CTX + *bla*P, *bla*TEM + *bla*SHV + *bla*DHA, *bla*TEM + *bla*IMP + *bla*KPC and *bla*CTX + *bla*SHV + *bla*BIL phenotypes. In total, 4.13% (*n* = 9) carried four members of beta-lactamase gene families, including the *bla*TEM + *bla*CTX + *bla*SHV + *bla*BIL, *bla*TEM + *bla*CMY + *bla*VIM + *bla*SHV, *bla*TEM + *bla*CTX +*bla*DHA + *bla*BIL, *bla*TEM + *bla*CTX + *bla*SHV + *bla*IMP, *bla*SHV + *bla*CTX + *bla*CMY + *bla*DHA and *bla*TEM + *bla*CTX + *bla*BIL + *bla*P phenotypes. In total, 21.56% (*n* = 47) of all *Escherichia coli* strains carried from five to nine different gene families, which formed uncommon phenotypes (Table 7).

### 3.8. Beta-Lactamase Genotypes

Whole-genome sequencing using NGS was carried out to determine the specific genotypes for the beta-lactamase genes. For the *bla*TEM genotypes (*n* = 209), we found that they were composed of one of the following genes: *bla*TEM-1 (*n* = 116, 55.5%), *bla*TEM-2 (*n* = 21, 10.05%), *bla*TEM-10 (*n* = 9, 4.31%), *bla*TEM-12 (*n* = 29, 13.88%), *bla*TEM-24 (*n* = 15, 7.18%) and *bla*TEM-52. For the *bla*CTX genotypes (*n* = 155), we found that they were composed of one of the following genes: *bla*CTX-M-1 (*n* = 28, 18.06%), *bla*CTX-M-15 (*n* = 106, 68.39%), *bla*CTX-M-27 (*n* = 7, 4.52%) and *bla*CTX-M100 (*n* = 14, 9.03%). For the blaDHA genotypes (*n* = 42), we found that they were composed of one of the following genes: *bla*DHA-1 (*n* = 34, 80.95%), *bla*DHA-2 (*n* = 6, 14.29%) and *bla*DHA-7 (*n* = 2, 4.76%). For the *bla*LAP genotype (*n* = 15), we found that it was composed of one of the following genes: *bla*LAP-1 (*n* = 11, 73.33%) and blaLAP-2 (*n* = 4, 26.67%). For the blaSHV genotype (*n* = 74), we found that it was composed of one of the following genes: *bla*SHV-1 (*n* = 7, 9.46%), *bla*SHV-11 (*n* = 2, 2.70%), *bla*SHV-12 (*n* = 58, 78.38%), *bla*SHV-100 (*n* = 4, 5.41%) and *bla*SHV-121 (*n* = 3, 4.05%). For the *bla*CMY genotype (*n* = 41), we found that it was composed of one of the following genes: *bla*CMY-2 (*n* = 37, 90.24%), *bla*CMY-9 (*n* = 1, 2.44%), *bla*CMY-12 (*n* = 1, 2.44%) and *bla*CMY-38 (*n* = 2, 4.88%). For the *bla*IMP genotypes (*n* = 24), we found that they were composed of one of the following genes: *bla*IMP-1 (*n* = 2, 8.33%), *bla*IMP-2 (*n* = 1, 4.17%), *bla*IMP-4 (*n* = 1, 4.17%), *bla*IMP-6 (*n* = 1, 4.17%), *bla*IMP-11 (*n* = 2, 8.33%) and *bla*IMP-14 (*n* = 17, 70.83%). For the *bla*KPC genotypes (*n* = 2), we found that they were composed of *bla*KPC-2 (*n* = 1, 50%) and *bla*KPC3 (*n* = 1, 50%). For the *bla*BIL (*n* = 65), *bla*P (*n* = 10), *bla*VIM (*n* = 9) and *bla*CTX (*n* = 4) genotypes, we found only one member of each of these families, specifically the *bla*BIL-1, *bla*P, *bla*VIM-1, *bla*KPC-3 and *bla*CTX-M-15 genes (Table 6).

## 4. Discussion

*Escherichia coli* is a Gram-negative bacteria that has coexisted with humans since ancient times. It is one of the best-adapted microorganisms to the human environment and has become a member of the intestinal microbiota. It has a worldwide distribution and is known for its high genetic plasticity, which enables it to easily adapt to different environments. In our research, we found that 22.48% of all multi-resistant *Escherichia coli* were cultured from urine, 21.10% from vaginal discharge, 6.42% from catheter tips, 5.96% from blood cultures, 5.50% from feces cultures, 5.05% from sputum, 3.67% from biopsies and 0.92% from cerebrospinal fluid. Additionally, 28.90% were isolated from diverse discharge. These findings are consistent with previous reports where *Escherichia coli* was isolated from patients with nosocomial infections, urine infections, gut infections, neonatal meningitis, pneumonia, cholecystitis, peritonitis, osteomyelitis, infectious arthritis and even otitis. Moreover, *Escherichia coli* is the most frequent cause of bacteremia [28,29,30,31,32,33,34,35,36,37].

### 4.1. *Escherichia coli* Nosocomial Characterization

*Escherichia coli* has been identified as a member of the nosocomial bacteriome in hospitals worldwide [38]. In our study, we determined the nosocomial origin of *Escherichia coli* by quantifying the multiple-antibiotic-resistance index (MAR index) using dilution antibiogram data (Table 3). Our analysis revealed a wide range of antibiotic resistance in the *Escherichia coli* strains obtained from in-hospital patients, with 99.08% having an MAR index greater than 0.25 and a mean of 0.58 ± 0.21 for the antibiotics conforming to the general dilution antibiogram (Table 2). For the beta-lactam selection antibiogram, the MAR index was 0.82 ± 0.09 and for both sets of antibiotics, the MAR index was larger than 4, indicating that all *Escherichia coli* strains were isolated from an environment of high antibiotic use, specifically nosocomial sources.

Our findings are consistent with other reports where *Escherichia coli* isolated from tertiary hospitals in South-West Nigeria had an MAR index higher than 0.2 [39]. Similar results were found in *Escherichia coli*-causing urine infections in patients from a referral hospital in Eastern Nepal [40]. The high MAR index is a major health problem caused by uncontrolled antibiotic use worldwide, especially in the nosocomial environment. It has been identified as a risk factor for neonatal mortality and prolonged hospital stays [41]. Similar results have been published in adult in-hospital patients infected with Gram-negative bacilli that have a high MAR index, with these patients showing a significant increase in mortality, morbidity, length of hospitalization and healthcare costs [42].

### 4.2. *Escherichia coli* Characterization and Genome Size

The metagenomic DNA analysis of sequencing data confirmed that the *Escherichia coli* strains obtained from the microbiology section of the clinic laboratory of the “October first” Regional Hospital had a 100% identity (Table 5). All strains showed at least 60% of all sequencing reads that identified the species, with a mean of 85% of sequencing reads. These findings are consistent with previous reports by our research group [20].

The genome sequence revealed three distinct groups (Figure 1) classified by size in base pairs: a short genome size composed of sequences from 4.56 to 4.83 Mbp (*n* = 10, 4.59%), a standard genome size composed of sequences from 4.84 to 5.27 Mbp (*n* = 176, 80.73%) and a long genome size composed of sequences from 5.28 to 6.07 Mbp (*n* = 32, 14.68%). These genome sizes are consistent with previous data published on the 61 genomes of *Escherichia coli* [43] and the data bank published by the SRA of NCBI [44].

### 4.3. Beta-Lactam Gene Families

The isolated *Escherichia coli* strains were cultured and selected in the presence of beta-lactam drugs in liquid culture media (Table 4). All strains were expected to be carriers of at least one member of the different beta-lactamase gene families. Real-time PCR was used to identify the gene families transcribed and the phenotype responsible for beta-lactam resistance. The results showed that 95.87% of the isolated strains were positive for *bla*TEM, 71.10% were positive for *bla*CTX, 33.94% were positive for *bla*SHV, 29.82% were positive for *bla*BIL, 19.27% were positive for *bla*DHA, 18.81% were positive for *bla*CMY, 11.01% were positive for *bla*IMP, 6.88% were positive for *bla*LAP, 14.59% were positive for *bla*P, 4.13% were positive for *bla*VIM, 1.82% were positive for *bla*CTX and 0.92% were positive for both *bla*KPC and *bla*CTX (Table 8). No *Escherichia coli* strains positive for members of the *bla*OXA and *bla*ROB gene families were found in all obtained strains from patients with nosocomial infections. In this study, the *bla*TEM gene family was detected in 95.87% of the studied *Escherichia coli* bacteria from patients diagnosed with nosocomial infection in the“October first” Regional Hospital. This result is consistent with previous reports that found the *bla*TEM gene family in 74.5% of all multi-drug-resistant ESBL-producing *Escherichia coli* isolates from hospitals in Malaysia [45]. The seven most frequent gene families that code for beta-lactamases expressed by *Escherichia coli* that cause nosocomial infections are *bla*TEM, *bla*CTX, *bla*SHV, *bla*BIL, *bla*DHA, *bla*CMY and *bla*IMP, representing 93.85% of the beta-lactamase problem. However, the *bla*TEM, *bla*CTX and *bla*SHV families together represent the greatest problem with regard to resistance to beta-lactam drugs, as they are expressed in 67.38% of all multi-resistant *Escherichia coli* strains that cause nosocomial infections in the in-patients of the “October first” Regional Hospital of the ISSSTE. These results are consistent with previous works that found that *Escherichia coli* isolated from in-hospital patients expressed different members of the *bla*CTX, *bla*SHV and *bla*TEM gene families [46,47,48,49,50,51]. On the other hand, the gene families *bla*BIL, *bla*DHA, *bla*CMY, *bla*IMP, *bla*LAP, *bla*P, *bla*VIM, *bla*CTX and *bla*KPC represent 32.62% of the beta-lactam antibiotic resistance problem. These results are consistent with previous reports where the expression of the following gene families in *Escherichia coli* isolated from in-hospital patients was found: *bla*BIL [52], *bla*DHA [53], *bla*CMY [54], *bla*IMP, *bla*LAP, which has been associated with quinolone resistance gene expression [55] and *bla*P [56].

### 4.4. Beta-Lactamase Genome Family Resistome

Out of the 218 *Escherichia coli* strains resistant to beta-lactam drugs, isolated from 351 in-hospital patients with nosocomial infections, a total of 52 different combinations of gene families coding for beta-lactamases were identified as contributing to the beta-lactam resistome. Interestingly, 13 of the most frequently occurring combinations accounted for a substantial 72.9% of all beta-lactam resistomes (Table 6). On the other hand, the remaining 40 less common combinations were responsible just for 27.1% of all multi-resistance to beta-lactam drugs (Table 7). The common combinations consisted of one to six distinct gene families that encode beta-lactamases (Table 6), whereas the less common combinations were composed of two to nine different gene families coding for beta-lactamases (Table 7).

These observations lead us to postulate that the rare combinations represent natural trials in which beta-lactam resistomes are formed by integrating a large number of beta-lactamase gene families. These combinations are likely to be subjected to selection pressure and might potentially evolve into highly efficient resistomes in the future. However, currently, the most successful resistomes are the common ones composed of six or fewer gene families, as they are prevalent in the majority of nosocomial infections caused by multi-resistant *Escherichia coli* strains.

### 4.5. Genomic Beta-Lactamase Intrafamilial Variability

#### 4.5.1. *bla*TEM Gene Family

The sequencing data revealed that 95.87% of all resistant *Escherichia coli* strains carried a member of the *bla*TEM gene family, with 21.56% of them exhibiting a unique beta-lactam resistance mechanism. In contrast, 74.31% of the strains (Table 6) showed multiple beta-lactam resistance mechanisms. These findings align with previous reports where the *bla*TEM gene family (*n* = 61 strains, 100%) was identified as the primary beta-lactam drug resistance mechanism in *Escherichia coli* isolated from the river Yamuna [57]. Characterization of the blaTEM members identified six different alleles of this gene family, including blaTEM-1 expressed in 55.5% of all isolated *Escherichia coli* strains, followed by *bla*TEM-12 in 13.88%, *bla*TEM-2 in 10.05%, *bla*TEM-52 in 9.09%, *bla*TEM-24 in 7.18% and *bla*TEM-10 in 4.31% (Table 6). These findings are consistent with another study where 24% (*n* = 48) of strains obtained from five hospitals in Tehran were found to carry the *bla*TEM-1 gene [58]. The blaTEM-1 enzyme is particularly noteworthy as it hydrolyzes ampicillin at a higher rate compared to carbenicillin, oxacillin and cephalothin but does not hydrolyze extended-spectrum cephalosporins. Furthermore, *bla*TEM-1 is considered an ancestral gene for *bla*TEM-10 and *bla*TEM-12, which are among the most commonly encountered *blaTEM* alleles in the United States of America [59].

Additionally, the *bla*TEM-2 allele exhibits similar hydrolytic activity to *bla*TEM-1 but possesses greater transcriptional activity [60]. Another less common beta-lactamase identified in this study includes *bla*TEM-24 and *bla*TEM-52, both of which have been described in bacteria-causing nosocomial infections in a Tunisian hospital [61]. These findings are consistent with similar results published by a Korean group from Clinical Isolates of *Escherichia coli* expressing the *bla*TEM-1, *bla*TEM-19, *bla*TEM-20 and *bla*TEM-52 genes [62], conferring resistance characteristics to penicillins, aztreonam and cephalosporins. The expression of all these genes has been previously cloned, sequenced and reported in *Escherichia coli* strains [63].

#### 4.5.2. *bla*CTX Gene Family

The second most frequent gene family identified in our study was *blaCTX* (54.72%), which was expressed in six isoforms, distributed across three clusters. Within the *bla*CTX-M-1 cluster, we found two members: *bla*CTX-M-1, present in 18.06% and *bla*CTX-M-15, found in 68.39% of the samples. In the *bla*CTX-9 cluster, only the*bla*CTX-M-27 was detected in 4.52% of the strains. The *bla*CTX-M-25 cluster had the *bla*CTX-M-100 isoform as its sole member, accounting for 9.03% of the isolates. Notably, the *bla*CTX-M-15 isoform emerged as the most dominant among the clusters. These findings are consistent with results obtained from clinically significant *Escherichia coli* isolates in Kuwait Hospitals, where the *bla*CTX-M-15 isoform was present in 84.1% of all isolates (Table 6) [64]. Additionally, another study conducted in the UK using *Escherichia coli*-producing CTX-M extended-spectrum beta-lactamases identified the *bla*CTX-M-15 isoform in 95.9% of all studied strains [65]. In contrast, our study found the blaCTX-M-1 isoform in 18.06% of the samples. Interestingly, previous works published by three university hospitals in Tehran reported that 61.8% of 144 CTX-resistant *Escherichia coli* isolates carried the *bla*CTX-M-1 gene [66], conferring resistance characteristics to penicillin, aztreonam and cephalosporin. The expression of all these genes has been previously cloned, sequenced and reported in *Escherichia coli* strains [67].

#### 4.5.3. *bla*SHV Gene Family

The third most frequently expressed beta-lactamase by *Escherichia coli* strains causing nosocomial infections was *bla*SHV, accounting for 33.94% of all analyzed multi-resistant strains. Within this gene family, five isoforms were identified, with *bla*SHV-12 being the most dominant (78.38%), followed by *bla*SHV-1 (9.46%), *bla*SHV-100 (5.41%), *bla*SHV-121 (4.05%) and *bla*SHV-11 (2.70%) (Table 6). Although all isoforms were expressed at low frequencies, *bla*SHV-12 stood out as the most prevalent member, isolated from diverse sources in a recent study conducted in Catalonia, where it was found at a frequency of 23% [68]. These isoforms confer characteristic resistance to penicillin, aztreonam and cephalosporin. The expression of all these gene isoforms has been previously cloned, sequenced and reported in *Escherichia coli* strains [69].

#### 4.5.4. *bla*IMP Gene Family

The blaIMP gene family was detected in 11.01% of the strains and it comprises six different members, namely *bla*IMP-1 (8.33%; *n* = 2), *bla*IMP-2 (4.17%; *n* = 1), *bla*IMP-4 (4.17%; *n* = 1), *bla*IMP-6 (4.17%, *n* = 1), *bla*IMP-11 (8.33%; *n* = 2) and the dominant gene *bla*IMP-14 (8.33%; *n* = 17). These genes confer characteristic resistance to imipenem and carbapenem [55]. The expression of all these genes has been previously cloned and sequenced from *Escherichia coli* strains [70].

#### 4.5.5. *bla*CTX-M, *bla*VIM and *bla*P Gene Family

All of these beta-lactamase gene families were expressed in only one isoform each. Specifically, the *bla*CTX-M-15 isoform was found in 1.82% (*n* = 4) of the strains, *bla*VIM-1 in 4.13% (*n* = 9) and *bla*P in 6.88% (*n* = 15) of the 218 *Escherichia coli* strains studied, isolated from patients attending the “October first” Regional Hospital in Mexico City. These specific isoforms of *bla*CTX-M-15, *bla*VIM-1 and *bla*P beta-lactamases were detected in the given samples, which might explain the observed differences from their more commonly reported counterparts worldwide, like *bla*CTX-M-1 and *bla*CTX-M-15, which are frequently identified extended-spectrum beta-lactamases (ESBLs) globally. The prevalence of specific beta-lactamase isoforms could be influenced by the sample type and origin.

It is worth noting that both *bla*CTX-M-1, *bla*VIM-1 and *bla*P beta-lactamases have been previously cloned and sequenced in *Escherichia coli* strains [71,72,73].

#### 4.5.6. *bla*LAP and *bla*KPC Gene Families

The blaLAP gene was expressed in 6.88% of the isolated *Escherichia coli* strains, with only two detected isoforms: *bla*LAP-1 was the dominant gene expressed, accounting for 73.33% of all positive *bla*LAP strains, followed by the *bla*LAP-2 gene isoform, representing 26.67%. This gene, in combination with other beta-lactamases, contributes to the multi-drug resistance characteristics [74].

Similarly, the *bla*KPC gene family was expressed in two isoforms, namely *bla*KPC-2 and *bla*KPC-3, both of which have been previously reported as members of the *Escherichia coli* genotype [75,76]. These isoforms provide carbapenem resistance characteristics [77].

#### 4.5.7. *bla*BIL, *bla*DHA and *bla*CMY Gene Families

The *bla*BIL gene family was the fourth most frequent, but only the *bla*BIL-1 isoform was identified. Previous reports have confirmed the expression of *bla*BIL-1 in *Escherichia coli*, conferring resistance characteristics to penicillin, carbenicillin, cefaclor and cephalosporin [52].

Additionally, the *bla*DHA (19.27%) and *bla*CMY (18.81%) gene families were expressed at similar frequencies. Within the *bla*DHA gene family, three isoforms were identified: *bla*DHA-1 (80.95%), *bla*DHA-2 (14.29%) and *bla*DHA-7 (4.76%). The *bla*CMY gene family was expressed in four isoforms: *bla*CMY-2 (90.24%), *bla*CMY-38 (4.88%), *bla*CMY-9 and *bla*CMY-12 (both expressed in 2.44%). The most dominant isoforms carried by *Escherichia coli*-causing nosocomial infections were *bla*DHA-1 and *bla*CMY-2. These genes were previously described as co-expressed with the *bla*CTX-M-15 isoform in a multi-resistant *Escherichia coli* strain causing the death of a puppy in Italy, indicating ESBL characteristics [78].

All of the blaBIL, blaDHA and blaCMY genes found in this study have been previously cloned and sequenced in *Escherichia coli* [52,79,80].

### 4.6. Beta-Lactam Resistome

The beta-lactam antibiotics, particularly penicillin, ampicillin and third-generation cephalosporin, play a crucial role in the treatment of nosocomial infections caused by *Escherichia coli* [81]. However, their extensive use in hospitals exerts significant selection pressure on the entire bacteriome, leading to the emergence of new beta-lactamases. This has become the most significant mechanism of resistance to beta-lactam antibiotics [82]. These resistance mechanisms are encoded in the bacterial chromosome, plasmids or transposons, facilitating the rapid spread of beta-lactamase genes among bacterial populations. As a result, the development of novel strategies to combat antibiotic resistance becomes essential to tackle the growing challenges posed by nosocomial infections.

In this study, our focus was on the identification of 14 of the most clinically important beta-lactamase families. These included the *bla*BIL, *bla*CFX, *bla*CMY, *bla*CTX, *bla*DHA, *bla*IMP, *bla*KPC, *bla*LAP, *bla*OXA, *bla*P, *bla*ROB, *bla*SHV, *bla*TEM and *bla*VIM gene families, collectively covering 99% of the beta-lactam resistance reported in *Escherichia coli* strains causing various intestinal, extraintestinal and systemic pathologies [83,84]. By focusing on these specific gene families, we aimed to gain valuable insights into the prevalence and diversity of beta-lactam-resistance mechanisms in clinically relevant *Escherichia coli* strains, which is crucial for understanding and combating antibiotic resistance in medical settings.

Among the *Escherichia coli* strains analyzed, we observed that they express from one to nine members of beta-lactamase gene families. It is worth noting that the accumulation of four or more members of the beta-lactamase gene families is a rare occurrence, as they are often unique strains. Nonetheless, these strains remain of utmost importance as they represent potential sources for the expansion of pan-resistant *Escherichia coli* strains with resistance to all beta-lactam drugs. The presence and diversity of these beta-lactamase genes emphasize the urgent need for effective measures to control and combat antibiotic resistance in nosocomial infections

In our study, the dominant beta-lactamase genotypes in the analyzed *Escherichia coli* strains were formed by *bla*TEM (21.6%) and its combinations with *bla*CTX (19.3%), *bla*CTX and *bla*SHV (6%) and *bla*CTX and *bla*BIL (5.5%). Together, these four genotypes constituted 52.4% of all identified genotypes. Based on these findings, the analyzed *Escherichia coli* strains are classified as extended-spectrum beta-lactamase-producing (ESBL) strains. The remaining combinations of beta-lactamase gene families represented from 0.5% to 4.1% each, but they still represent pan-resistant *Escherichia coli* strains against beta-lactam drugs. Our method of evaluation using real-time PCR and corroborated by next-generation sequencing provides a comprehensive panorama of the beta-lactam drug resistome in the analyzed *Escherichia coli* strains.

These results are in line with previous reports indicating that *Escherichia coli* is the bacterial species that most frequently produces ESBLs [85]. Similarly, in *Klebsiella pneumoniae*, it has been reported that multiple beta-lactamase gene carriers express from two to four different members of the beta-lactamase gene families [86]. However, it is essential to note that our method provides a broader and more generalized perspective of the beta-lactam drug resistome in the studied *Escherichia coli* strains, allowing for a comprehensive understanding of the resistance mechanisms.

The molecular methods used for the identification and characterization of the *Escherichia coli* beta-lactam resistome are of critical importance in drug therapy. In nosocomial infections caused by ESBL-producing *Escherichia coli*, the available drug therapy options are significantly reduced due to the pan-resistant phenotype. This can lead to an increase in mortality rates, prolonged hospitalization time and higher healthcare costs [87,88]. Patients who have acquired nosocomial infections caused by ESBL-producing *Escherichia coli* have been reported to face challenges in treatment [89]. For instance, the identification of *Escherichia coli* strains producing ESBLs of the CTX-M type has been linked to treatment failure in urinary-tract infections (UTIs) when treated with commonly used antibiotics such as penicillin, ampicillin and first- to fourth-generation cephalosporins. Even next-generation antibiotics like carbapenem and imipenem may prove ineffective against these strains, as some of them have also developed resistance to beta-lactamase inhibitors [90,91].

Our study’s results are consistent with those published in a multicenter study conducted in eight hospitals in Mexico, where the dominant isolated microorganisms were ESBL-producing *Escherichia coli* strains (83.9%) and many carried genes encoding beta-lactamases of *bla*SHV and/or *bla*CTX-M types [92]. These findings emphasize the urgent need for robust molecular methods to identify and characterize beta-lactam resistance in *Escherichia coli*, as they play a crucial role in guiding appropriate and effective drug therapy decisions for nosocomial infections.

## 5. Conclusions

The multi-resistance phenotype for beta-lactam drugs observed in *Escherichia coli*-causing nosocomial infections for in-hospital patients at different services within the “October first” Regional Hospital is primarily determined by genes belonging to six beta-lactamase gene families: *bla*TEM, *bla*CTX, *bla*SHV, *bla*BIL, *bla*DHA and *bla*CMY. These gene families, when combined, account for the most frequent phenotypes of the beta-lactam resistome, irrespective of the sample type and tissue source. Among these, the genotypes comprising the genes *bla*TEM1, *bla*CTX-M-15, *bla*SHV-12, *bla*BIL-1, *bla*DHA-1 and *bla*CMY-2 are found to be the most prevalent in the third-level hospital located in Mexico City. These findings provide valuable insights into the molecular basis of beta-lactam resistance in nosocomial *Escherichia coli* infections, which is essential for optimizing treatment strategies and controlling the spread of antibiotic-resistant strains in the hospital setting.

## Figures and Tables

**Figure 1 antibiotics-12-01355-f001:**
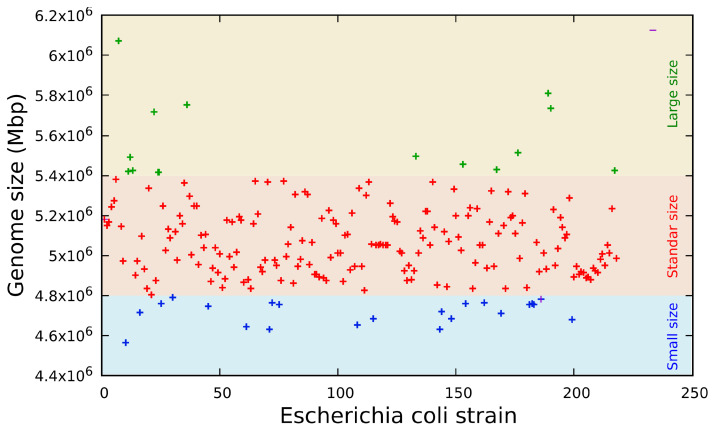
Genome size distribution of 218 *Escherichia coli* strains found to cause nosocomial infection, green points = large genomes, red points = standard genomes, blue points = small genomes.

**Figure 2 antibiotics-12-01355-f002:**
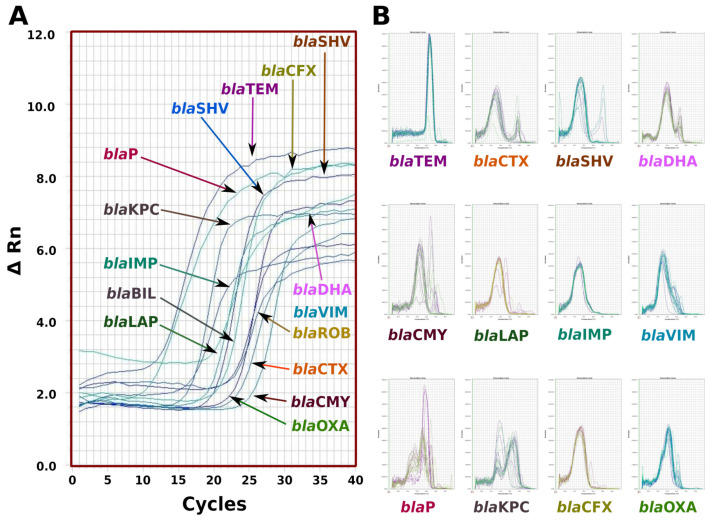
Characterization of the real-time PCR of the different families of genes encoding for beta-lactamases: (**A**) amplification curves, (**B**) dissociation curves. In both curves SYBR-GREEN was used as a fluorescent dye to show the qPCR amplification and dissociation.

**Table 1 antibiotics-12-01355-t001:** Primer sequences and amplicon size for beta-lactamase gene families [20].

Gene Family	Primer Name	Primer Sequence (5${}^{\prime }$ to 3${}^{\prime }$) *	Tm (${}^{\circ }$C)	Position	Amplicon (bp)
*bla*OXA	*bla*OXA-FW	GGTTTCGGTAATGCTGAAATTGG	61.18	214–236	114
	*bla*OXA-RW	GCTGTGTATGTGCTAATTGGGA	61.19	327–306	
*bla*VIM	*bla*VIM-FW	CGACAGTCARCGAAATTCC	61.39	105–123	133
	*bla*VIM-RW	CAATGGTCTSATTGTCCGTG	61.34	238–219	
*bla*SHV	*bla*SHV-FW1	CGTAGGCATGATAGAAATGGATC	61.04	133–155	106
	*bla*SHV-RW1	CGCAGAGCACTACTTTAAAGG	61.33	239–218	
	*bla*SHV-FW2	GCCTCATTCAGTTCCGTTTC	61.62	399–418	141
	*bla*SHV-RW2	CCATTACCATGAGCGATAACAG	61.22	540–518	
*bla*TEM	*bla*TEM-FW	GCCAACTTACTTCTGACAACG	61.8	1699–1719	213
	*bla*TEM-RW	CGTTTGGAATGGCTTCATTC	60.13	1912–1892	
*bla*IMP	*bla*IMP-FW1	GGAATAGARTGGCTTAAYTCTCG	60.92	319–332	183
	*bla*IMP-RW1	CYASTASGTTATCTKGAGTGTG	62.45	502–480	
	*bla*IMP-FW2	GGTGGAATAGARTGGCTTAAYTC	61.11	316–339	192
	*bla*IMP-RW2	CCAAACCACTACGTTATCTKGAG	61.29	508–485	
*bla*ROB	*bla*ROB-FW	CCAACATCGTGGAAAGTGTAG	61.27	718–739	126
	*bla*ROB-RW	GTAAATTGCGTACTCATGATTGC	60.9	844–821	
*bla*KPC	*bla*KPC-FW	GCTAAACTCGAACAGGACTTTG	61.79	100–121	117
	*bla*KPC-RW	CTTGAATGAGCTGCACAGTG	61.9	216–197	
*bla*CTX	*bla*CTX-FW1	GATACCGCAGATAATACGCAG	60.79	161–181	116
	*bla*CTX-RW1	CGTTTTGCGTTTCACTCTG	60.28	276–258	
	*bla*CTX-FW2	GCTGATTCTGGTCACTTACTTC	61.02	789–810	83
	*bla*CTX-RW2	CGCCGACGCTAATACATC	60.69	855–872	
	*bla*CTX-FW3	CTGCTTAACTACAATCCSATTGC	62.17	314–336	226
	*bla*CTX-RW3	GGAATGGCGGTATTKAGC	60.86	539–522	
*bla*CMY	*bla*CMY-FW1	GTTTGAGCTAGGATCGGTTAG	60.25	337–357	123
	*bla*CMY-RW1	CTGTTTGCCTGTCAGTTCTG	61.48	460–441	
	*bla*CMY-FW2	GAACGAAGGCTACGTAGCT	61.71	213–231	160
	*bla*CMY-RW2	CTGAAACGTGATTCGATCATCA	61.08	372–351	
*bla*DHA	*bla*DHA-FW1	GCATATTGATCTGCATATCTCCAC	61.6	399–422	200
	*bla*DHA-RW1	GCTGCTGTAACTGTTCTGC	61.62	598–580	
	*bla*DHA-FW2	GCGGATCTGCTGAATTTCTATC	61.54	464–485	147
	*bla*DHA-RW2	GCAGTCAGCAACTGCTCATAC	61.05	610–591	
	*bla*DHA-FW3	GTAAGATTCCGCATCAAGCTG	61.74	430–450	117
	*bla*DHA-RW3	GGGTTATCTCACACCTTTATTACTG	61.08	546–522	
*bla*P	*bla*P-FW	GGAGAATATTGGGATTACAATGGC	61.74	271–294	204
	*bla*P-RW	CGCATCATCGAGTGTGATTG	61.8	474–455	
*bla*CFX	*bla*CFX-FW	CCAGTCATATCATTGACAGTGAG	60.86	437–459	177
	*bla*CFX-RW	GACATTTCCTCTTCCGTATAAGC	61.16	613–591	
*bla*LAP	*bla*LAP-FW	AGGGCTTGAACAACTTGAAC	61.07	249–268	126
	*bla*LAP-RW	GTAATGGCAGCATTGCATAAC	60.59	374–354	
*bla*BIL	*bla*BIL-FW	GCCGATATCGTTAATCGCAC	61.65	100–119	128
	*bla*BIL-RW	GTTATTGGCGATATCGGCTTTA	60.98	227–206	

* All primers were designed to run at the same temperature.

**Table 2 antibiotics-12-01355-t002:** Representative antibiotic-resistance pattern of *Escherichia coli* causal of nosocomial infection in in-hospital patients in the “October first” Regional Hospital (ISSSTE) in Mexico city.

Sample ID	Antibiotic Drugs																			
	AM	AN	SAM	ATM	CZ	FEP	CAN	CME	CRO	CXI	CIP	GM	IEM	MEM	PTZ	TCA	TM	STX	FT	GAN
21.242.01	R	S	R	R	R	R	R	R	R	S	R	R	S	S	R	S	S	R	S	S
29.269.39	R	S	R	R	R	R	R	R	R	R	R	S	S	S	R	S	S	S	S	R
16.231.77	R	S	R	R	R	R	S	R	R	S	R	R	S	S	S	R	R	S	S	R
14.074.98	R	S	R	R	R	R	R	R	R	R	R	S	S	S	R	S	S	R	S	S
16.242.17	R	S	R	R	R	R	S	R	R	R	R	R	S	S	S	S	R	R	S	R
9.312.99	R	S	R	R	R	R	R	R	R	S	R	R	R	S	R	S	R	R	S	S
25.012.34	R	S	R	R	R	R	S	R	R	R	R	S	S	S	S	S	R	R	S	R
25.275.37	R	S	R	R	R	R	R	R	R	S	R	R	S	S	R	S	S	R	S	R
30.318.09	R	S	R	R	R	R	R	R	R	R	S	S	S	S	R	R	S	R	S	S
23.089.11	R	S	R	R	R	R	S	R	R	R	R	R	S	S	S	S	R	R	R	R
30.343.91	R	S	R	R	R	R	R	R	R	S	R	S	S	R	R	S	S	R	S	R
13.223.15	R	S	R	R	R	R	S	R	R	R	R	R	S	S	S	S	R	R	R	S
13.330.01	R	S	R	R	R	R	R	R	R	R	R	S	S	S	R	S	S	R	S	S
15.286.98	R	R	R	R	R	R	R	R	R	R	R	S	S	S	S	S	R	R	S	R
14.273.94	R	S	R	R	R	R	R	R	R	S	R	S	R	R	R	S	S	R	S	R
8.204.99	R	R	R	R	R	R	S	R	R	R	R	S	S	S	S	S	R	R	S	S
14.001.99	R	R	R	R	R	R	S	R	R	S	R	S	S	R	R	S	R	R	R	R
3.296.34	R	S	R	R	R	R	R	R	R	R	R	R	S	S	R	S	S	R	S	S
21.045.37	R	R	R	R	R	R	R	R	R	S	R	S	S	R	S	R	R	R	S	R
20.315.10	R	S	R	R	R	R	R	R	R	R	R	S	S	R	R	S	S	R	S	R
3.286.91	R	S	R	R	R	R	R	R	R	R	R	S	R	R	S	S	R	R	S	R

**R** = resistant, **S** = sensitive, **AM** = ampicillin, **AN** = amikasin, **SAM** = ampicillin/sulbactam, **ATM** = aztreonam,
**CZ** = cefazolin, **FEP** = cefepime, **CAN** = cefotetan, **CME** = ceftazidime, **CRO** = ceftriaxone, **CXI** = cefuroxime,
**CIP** = ciprofloxacin, **GM** = gentamicin, **IEM** = imipenem, **MEM** = meropenem, **PTZ** = piperacillin/tazobactam,
**TCA** = ticarcillin/clavulanic acid, **TM** = tobramycin, **STX** = trimethoprim/sulfamethoxasol, **FT** = nitrofurantoin
and **GAN** = gatifloxacin.

**Table 3 antibiotics-12-01355-t003:** Multiple-antibiotic-resistance indices (MAR indices) of *Escherichia coli* strains causing nosocomial infection in the “October first” Regional Hospital (ISSSTE) in Mexico city.

Number of Antibiotics	MAR Index *	Frequency	Number of Antibiotics	MAR Index	Frequency
1	0.05	0 (0.00%)	11	0.55	10 (4.59%)
2	0.10	0 (0.00%)	12	0.60	32 (14.68%)
3	0.15	0 (0.00%)	13	0.65	34 (15.60%)
4	0.20	0 (0.00%)	14	0.70	26 (11.93%)
5	0.25	2 (0.92%)	15	0.75	37 (16.97%)
6	0.30	5 (2.29%)	16	0.80	13 (5.96%)
7	0.35	10 (4.59%)	17	0.85	5 (2.29%)
8	0.40	9 (4.13%)	18	0.90	3 (1.38%)
9	0.45	13 (5.96%)	19	0.95	0 (0.00%)
10	0.50	19 (8.72%)	20	1.00	0 (0.00%)

* The MAR index was calculated as a function of all beta-lactam antibiotics used in the antibiogram.

**Table 4 antibiotics-12-01355-t004:** Frequency of *Escherichia coli* * coli multi-resistant strains to beta-lactam antibiotics in the “October first” Regional Hospital (ISSSTE) in Mexico city.

Beta-Lactam Antibiotics	*Escherichia coli*	
	Frequency	%
Amoxicillin/clavulanic acid (AmC-30)	190	87.16
Piperacillin (PIP-100)	141	64.68
Piperacillin/tazobactam (TZP-110)	39	17.89
Doxycycline (D-30)	114	52.29
Ampicillin 10(AM-10)	218	100
Cephalothin 1${}^{\circ }$ (CF-30)	218	100
Cefazolin 1${}^{\circ }$ (CZ-30)	218	100
Cefaclor 2${}^{\circ }$ (CEC-30)	147	67.43
Cefuroxime 2${}^{\circ }$ (CXM-30)	158	72.48
Cefixime 3${}^{\circ }$ (CFM-5)	186	85.32
Ceftizoxime 3${}^{\circ }$ (ZOX-30)	34	15.6
Meropenem (MEM-10)	13	28.34
Imipenem (IPM-10)	24	11.01

* All strands were cultured at the Microbiology Laboratory of the “October first” Regional Hospital.

**Table 5 antibiotics-12-01355-t005:** Frequency of beta-lactamase gene families identified by real-time PCR in multi-resistant *Escherichia coli* strains causing nosocomial infection in the “October first” Regional Hospital.

Gene Family	Positive Strains (*n*)	%	Gene Family	Positive Strains (*n*)	%
*bla*TEM	209	95.87	*bla*IMP	24	11.01
*bla*CTX	155	71.1	*bla*LAP	15	6.88
*bla*SHV	74	33.94	*bla*P	10	4.59
*bla*BIL	65	29.82	*bla*VIM	9	4.13
*bla*DHA	42	19.27	*bla*CTX	4	1.82
*bla*CMY	41	18.81	*bla*KPC	2	0.92

**Table 6 antibiotics-12-01355-t006:** Uncommon beta-lactam resistomes carried by multi-resistant *Escherichia coli* strains causing nosocomial infection in the “October first” Regional Hospital (ISSSTE) in Mexico city.

Beta-Lactam Gene Family Phenotype	*n*	%
*bla*TEM + *bla*SHV + *bla*CTX	2	0.9
*bla*TEM + *bla*CTX + *bla*SHV + *bla*DHA + *bla*CMY + *bla*LAP	2	0.9
*bla*SHV + *bla*DHA	2	0.9
*bla*SHV + *bla*CTX + *bla*BIL	2	0.9
*bla*TEM + *bla*CMY	1	0.5
*bla*TEM + *bla*LAP	1	0.5
*bla*CTX + *bla*BIL	1	0.5
*bla*TEM + *bla*SHV + *bla*BIL	1	0.5
*bla*TEM + *bla*CTX + *bla*P	1	0.5
*bla*TEM + *bla*SHV + *bla*BIL	1	0.5
*bla*TEM + *bla*CTX + *bla*DHA	1	0.5
*bla*TEM + *bla*CTX + *bla*IMP	1	0.5
*bla*TEM + *bla*CFX + *bla*SHV	1	0.5
*bla*TEM + *bla*CMY + *bla*BIL	1	0.5
*bla*TEM + *bla*CMY + *bla*P	1	0.5
*bla*TEM + *bla*SHV + *bla*DHA	1	0.5
*bla*TEM + *bla*IMP + *bla*KPC	1	0.5
*bla*CTX + *bla*SHV + *bla*BIL	1	0.5
*bla*TEM + *bla*CMY + *bla*VIM + *bla*SHV	1	0.5
*bla*TEM + *bla*CTX + *bla*DHA + *bla*BIL	1	0.5
*bla*TEM + *bla*CTX + *bla*SHV + *bla*IMP	1	0.5
*bla*SHV + *bla*CTX + *bla*CMY + *bla*DHA	1	0.5
*bla*TEM + *bla*CTX + *bla*BIL + *bla*P	1	0.5
*bla*TEM + *bla*CTX + *bla*SHV + *bla*CMY + *bla*VIM	1	0.5
*bla*TEM + *bla*CTX + *bla*SHV + *bla*CMY + *bla*DHA	1	0.5
*bla*TEM + *bla*CTX + *bla*SHV + *bla*BIL + *bla*P	1	0.5
*bla*TEM + *bla*CTX + *bla*CMY + *bla*IMP + *bla*DHA	1	0.5
*bla*TEM + *bla*CTX + *bla*DHA + *bla*IMP + *bla*BIL + *bla*KPC	1	0.5
*bla*TEM + *bla*CTX + *bla*CFX + *bla*DHA + *bla*BIL + *bla*P	1	0.5
*bla*TEM + *bla*CTX + *bla*SHV + *bla*IMP + *bla*DHA + *bla*BIL	1	0.5
*bla*TEM + *bla*CTX + *bla*SHV + *bla*CMY + *bla*DHA + *bla*BIL	1	0.5
*bla*TEM + *bla*CTX + *bla*SHV + *bla*IMP + *bla*DHA + *bla*BIL	1	0.5
*bla*TEM + *bla*CTX + *bla*CFX + *bla*SHV + *bla*IMP + *bla*BIL	1	0.5
*bla*TEM + *bla*CTX + *bla*SHV + *bla*CMY + *bla*DHA + *bla*P	1	0.5
*bla*TEM + *bla*CTX + *bla*SHV + *bla*IMP + *bla*DHA + *bla*BIL + *bla*P	1	0.5
*bla*TEM + *bla*CTX + *bla*SHV + *bla*CFX + *bla*IMP + *bla*BIL + *bla*P	1	0.5
*bla*TEM + *bla*CTX + *bla*CMY + *bla*SHV + *bla*IMP + *bla*DHA + *bla*BIL + *bla*P	1	0.5
*bla*TEM + *bla*CTX + *bla*SHV + *bla*CFX + *bla*IMP + *bla*DHA + *bla*BIL + *bla*LAP	1	0.5
*bla*TEM + *bla*CTX + *bla*CMY + *bla*SHV + *bla*IMP + *bla*BIL + *bla*DHA + *bla*P + *bla*BIL	1	0.5

**Table 7 antibiotics-12-01355-t007:** Beta-lactamase-specific genes expressed by multi-resistant *Escherichia coli* strains causing nosocomial infection in the “October first” Regional Hospital (ISSSTE) in Mexico city.

Gene Family	Genotype	Frequency	%	Gene Family	Genotype	Frequency	%
*bla*TEM (*n* = 209)	*bla*TEM-1	116	55.5	*bla*SHV (*n* = 74)	*bla*SHV-1	7	9.46
	*bla*TEM-2	21	10.05		*bla*SHV-11	2	2.7
	*bla*TEM-10	9	4.31		*bla*SHV-12	58	78.38
	*bla*TEM-12	29	13.88		*bla*SHV-100	4	5.41
	*bla*TEM-24	15	7.18		*bla*SHV-121	3	4.05
	*bla*TEM-52	19	9.09	*bla*BIL (*n* = 65)	*bla*BIL-1	65	100
*bla*CTX (*n* = 155)	*bla*CTX-M-1	28	18.06	*bla*CMY (*n* = 41)	*bla*CMY-2	37	90.24
	*bla*CTX-M-15	106	68.39		*bla*CMY-9	1	2.44
	*bla*CTX-M-27	7	4.52		*bla*CMY-12	1	2.44
	*bla*CTX-M100	14	9.03		*bla*CMY-38	2	4.88
*bla*DHA (*n* = 42)	*bla*DHA-1	34	80.95	*bla*IMP (*n* = 24)	*bla*IMP-1	2	8.33
	*bla*DHA-2	6	14.29		*bla*IMP-2	1	4.17
	*bla*DHA-7	2	4.76		*bla*IMP-4	1	4.17
*bla*LAP (*n* = 15)	*bla*LAP-1	11	73.33		*bla*IMP-6	1	4.17
	*bla*LAP-2	4	26.67		*bla*IMP-11	2	8.33
*bla*P (*n* = 10)	*bla*P	10	100		*bla*IMP-14	17	70.83
*bla*VIM (*n* = 9)	*bla*VIM-1	9	100	*bla*KPC (*n* = 2)	*bla*KPC-2	1	50
*bla*CTX (*n* = 4)	*bla*CTX-M-15	4	100		*bla*KPC-3	1	50

All beta-lactamase genes were confirmed using NGS data.

**Table 8 antibiotics-12-01355-t008:** Frequent beta-lactam resistomes carried by multi-resistant *Escherichia coli* strains causing nosocomial infection in the “October first” Regional Hospital (ISSSTE) in Mexico city.

Beta-Lactam Gene Family Phenotype	*n*	%
*bla*TEM	47	21.6
*bla*TEM + *bla*CTX	42	19.3
*bla*TEM + *bla*SHV + *bla*CTX	13	6
*bla*TEM + *bla*CTX + *bla*BIL	12	5.5
*bla*TEM + *bla*CTX + *bla*CMY	8	3.7
*bla*TEM + *bla*CTX + *bla*SHV + *bla*CMY + *bla*BIL	8	3.7
*bla*TEM + *bla*CTX + *bla*SHV + *bla*DHA + *bla*BIL	7	3.2
*bla*TEM + *bla*IMP + *bla*SHV + *bla*DHA + *bla*BIL	6	2.8
*bla*CTX + *bla*VIM	4	1.8
*bla*TEM + *bla*CTX + *bla*SHV + *bla*BIL	4	1.8
*bla*TEM + *bla*SHV + *bla*CTX + *bla*CMY + *bla*BIL	4	1.8
*bla*TEM + *bla*CTX + *bla*CMY + *bla*SHV + *bla*IMP + *bla*BIL	4	1.8
*bla*TEM + *bla*CTX + *bla*VIM	3	1.4

## Data Availability

The data presented in this study are available from the corresponding author.

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
