# Peer review of "Characterization of Beta-Lactam Resistome of Escherichia coli Causing Nosocomial Infections"

_antibiotics, 2023, doi:10.3390/antibiotics12091355_

Round 1

Reviewer 1 Report

In this study, the authors conducted a comprehensive evaluation of the conformation of the beta-lactam resistome, which is responsible for the specific resistance pattern observed against beta-lactam antibiotics. They isolated a total of 218 Escherichia coli strains from various sources such as urine, vaginal discharge, catheter tips, feces, sputum, biopsies, cerebrospinal fluid, and other unspecified discharges of in-hospital patients with nosocomial infections. All strains underwent antimicrobial profiling using microdilution method and were cultured in beta-lactam supplemented Luria medium. The strains were then subjected to multiplex PCR and next-generation sequencing to characterize the beta-lactam resistome. The findings revealed that all strains exhibited a multidrug-resistant phenotype, particularly against beta-lactam drugs. The predominant gene family associated with this resistance pattern was blaTEM, which was expressed in 95.87% of the strains. Notably, the blaTEM gene was expressed either individually or combined with other genes such as blaCTX, blaSHV, and blaBIL. These results provide valuable insights into the genetic basis of beta-lactam resistance in Escherichia coli strains isolated from nosocomial infections.

In general, the manuscript would benefit from thorough revision to improve its English language usage. Additionally, there are areas that could be enhanced to elevate the quality and impact of the manuscript.

1.     Clarify the term "antibiotic microdilution" in line 8.

2.     In line 12, what is meant by "multiple convinations"?

3.     Please explain the meaning of "October first" in line 15, and differentiate between "October first" and "October firsth" as mentioned in line 83.

4.     Several sentences lack proper sentence structure formation, such as "results as a healthcare failure That conditions..." in line 21, "Although, the long in-hospital stay is..." in line 27, "were biochemestry identified by the..." in line 82, and "set ofh beta-lactam..." in line 125. Please correct these instances throughout the manuscript.

5.     It is recommended to italicize the isolate's name throughout the manuscript, including in lines 88, 103, and 220 (Figure 1 - X axis legend).

6.     The legend of Figure 2 needs to be rephrased, and it may be helpful to use different color coding to represent the amplification curve for better comprehension.

7.     In Figure 2 (b), additional peaks are observed in blaCTX, blaSHV, blaDHA, blaCMY, blaLAP, blaP, and blaKPC, which may indicate nonspecific binding.

Extensive editing of English language required

Author Response

In this study, the authors conducted a comprehensive evaluation of the conformation of the beta-lactam resistome, which is responsible for the specific resistance pattern observed against beta-lactam antibiotics. They isolated a total of 218 Escherichia coli strains from various sources such as urine, vaginal discharge, catheter tips, feces, sputum, biopsies, cerebrospinal fluid, and other unspecified discharges of in-hospital patients with nosocomial infections. All strains underwent antimicrobial profiling using microdilution method and were cultured in beta-lactam supplemented Luria medium. The strains were then subjected to multiplex PCR and next-generation sequencing to characterize the beta-lactam resistome. The findings revealed that all strains exhibited a multidrug-resistant phenotype, particularly against beta-lactam drugs. The predominant gene family associated with this resistance pattern was blaTEM, which was expressed in 95.87% of the strains. Notably, the blaTEM gene was expressed either individually or combined with other genes such as blaCTX, blaSHV, and blaBIL. These results provide valuable insights into the genetic basis of beta-lactam resistance in Escherichia coli strains isolated from nosocomial infections.

In general, the manuscript would benefit from thorough revision to improve its English language usage. Additionally, there are areas that could be enhanced to elevate the quality and impact of the manuscript.

1.     Clarify the term "antibiotic microdilution" in line 8.

According to the reviewer's observation, we have changed the word "microdilution" to "dilution" to avoid any confusion in the methodology.

2.     In line 12, what is meant by "multiple convinations"?

To clarify this confusion and according to the reviewer's suggestion, we have changed the words "multiple combination" to "various configurations."

3.     Please explain the meaning of "October first" in line 15, and differentiate between "October first" and "October firsth" as mentioned in line 83.

According to the reviewer's feedback, we have corrected the words "October firsth" to "October first."

4.     Several sentences lack proper sentence structure formation, such as "results as a healthcare failure That conditions..." in line 21, "Although, the long in-hospital stay is..." in line 27, "were biochemestry identified by the..." in line 82, and "set ofh beta-lactam..." in line 125. Please correct these instances throughout the manuscript.

To avoid lack of agreement in the sentences, a review of each paragraph was carried out. They were corrected and checked using the editing service provided by MDPI

5.     It is recommended to italicize the isolate's name throughout the manuscript, including in lines 88, 103, and 220 (Figure 1 - X axis legend).

According to the reviewer's suggestion, the name "Escherichia coli" was italicized throughout the document.

6.     The legend of Figure 2 needs to be rephrased, and it may be helpful to use different color coding to represent the amplification curve for better comprehension.

The amplification curves were directly obtained from the real-time thermocycler, and the color code is pre-set according to the position of the well. Therefore, we cannot assign a specific color to each tube, as, in general, 384 wells are run simultaneously, including the known controls for absolute quantification.

7. In Figure 2 (b), additional peaks are observed in blaCTX, blaSHV, blaDHA, blaCMY, blaLAP, blaP, and blaKPC, which may indicate nonspecific binding.

I fully agree with the comment; however, it is worth noting that this is what is expected from a qPCR performed with degenerate oligonucleotides. As stated in the manuscript, each tube contains at least 3 different sets of primers, making the amplifications specific for different members of each beta-lactamase family. Therefore, in graphs where multiple peaks are observed, this does not indicate non-specificity. Nevertheless, it is essential to observe that the majority of the observed peaks correspond to: 1. The beta-lactamase amplicon, 2. Identification of two isoforms of the same beta-lactamase family, and 3. The dissociation of dimers formed by the degenerate oligonucleotides

Comments on the Quality of English Language

The manuscript was submitted to the language editing service at MDPI, so this point is now resolved.

Extensive editing of English language required

According to the reviewer's comment, we have extensively edited the manuscript to make its reading more comprehensible.

Reviewer 2 Report

The main question addressed by the research is the characterization of antibiotic resistances in nosocomial infections caused by Escherichia coli. The study specifically focuses on the resistome of beta-lactams, which are a class of antibiotics commonly used to treat bacterial infections. The research aims to identify the genes responsible for beta-lactam resistance in nosocomial infection and to determine the prevalence and diversity of these genes in clinical isolates. The study also investigates the co-occurrence of beta-lactam resistance genes. Overall, the research seeks to provide a better understanding of the mechanisms of antibiotic resistance in E. coli and to inform the development of more effective treatment strategies for nosocomial infections caused by this pathogen.

The spread of nosocomial infections has become a major public health problem worldwide. Molecular epidemiological information on the frequencies of beta-lactam drug resistance determinants in hospital infections is always relevant.

The publication provides recent molecular epidemiological data on the determinants of beta-lactam resistance among nosocomial pathogens in Mexico.

The methodology is correct. It is gratifying that a holistic analysis of isolates was carried out in the work carried out. Genomic sequencing, biochemical tests and MIC determination were carried out. Additional questions about the methodology will be just nagging.

The article is descriptive, like much of epidemiology. The conclusions are well supported by the data presented.

The references are appropriate.

Tables and figures are accurate and sufficiently provide information.

There is one small remark.  Line 123 "Luria Bertani" is not the valid name for the LB medium.  Please correct it to "lysogeny broth". The developer of this medium Giuseppe Bertani in a later article (J Bacteriol. 2004 Feb; 186(3): 595-600. doi: 10.1128/JB.186.3.595-600.2004) insists that the abbreviation LB stands for lysogeny broth.

The article is well written. The amount of work done is great. I would like to wish success to the team.

Author Response

Comments and Suggestions for Authors

The main question addressed by the research is the characterization of antibiotic resistances in nosocomial infections caused by Escherichia coli. The study specifically focuses on the resistome of beta-lactams, which are a class of antibiotics commonly used to treat bacterial infections. The research aims to identify the genes responsible for beta-lactam resistance in nosocomial infection and to determine the prevalence and diversity of these genes in clinical isolates. The study also investigates the co-occurrence of beta-lactam resistance genes. Overall, the research seeks to provide a better understanding of the mechanisms of antibiotic resistance in E. coli and to inform the development of more effective treatment strategies for nosocomial infections caused by this pathogen.

The spread of nosocomial infections has become a major public health problem worldwide. Molecular epidemiological information on the frequencies of beta-lactam drug resistance determinants in hospital infections is always relevant.

The publication provides recent molecular epidemiological data on the determinants of beta-lactam resistance among nosocomial pathogens in Mexico.

The methodology is correct. It is gratifying that a holistic analysis of isolates was carried out in the work carried out. Genomic sequencing, biochemical tests and MIC determination were carried out. Additional questions about the methodology will be just nagging.

The article is descriptive, like much of epidemiology. The conclusions are well supported by the data presented.

The references are appropriate.

Tables and figures are accurate and sufficiently provide information.

There is one small remark.  Line 123 "Luria Bertani" is not the valid name for the LB medium.  Please correct it to "lysogeny broth". The developer of this medium Giuseppe Bertani in a later article (J Bacteriol. 2004 Feb; 186(3): 595-600. doi: 10.1128/JB.186.3.595-600.2004) insists that the abbreviation LB stands for lysogeny broth.

According to the reviewer's suggestion, we have changed the words "Luria Bertani" to "lysogeny broth."

The article is well written. The amount of work done is great. I would like to wish success to the team.

Thank you

Round 2

Reviewer 1 Report

The authors have made significant improvements to the manuscript in their revised version, and I recommend its publication in its current form.